# A Combined Deep Eutectic Solvent–Ionic Liquid Process for the Extraction and Separation of Platinum Group Metals (Pt, Pd, Rh)

**DOI:** 10.3390/molecules26237204

**Published:** 2021-11-27

**Authors:** Olga Lanaridi, Sonja Platzer, Winfried Nischkauer, Andreas Limbeck, Michael Schnürch, Katharina Bica-Schröder

**Affiliations:** 1Institution of Applied Synthetic Chemistry, Technische Universität Wien, 1060 Vienna, Austria; olga.lanaridi@tuwien.ac.at (O.L.); sonja.platzer@uniwie.ac.at (S.P.); michael.schnuerch@tuwien.ac.at (M.S.); 2Institute of Chemical Technologies and Analytics, Technische Universität Wien, 1060 Vienna, Austria; winfried.n@gmx.at (W.N.); andreas.limbeck@tuwien.ac.at (A.L.)

**Keywords:** ionic liquids, deep eutectic solvents, platinum group metals, recycling, extraction

## Abstract

Recovery of platinum group metals from spent materials is becoming increasingly relevant due to the high value of these metals and their progressive depletion. In recent years, there is an increased interest in developing alternative and more environmentally benign processes for the recovery of platinum group metals, in line with the increased focus on a sustainable future. To this end, ionic liquids are increasingly investigated as promising candidates that can replace state-of-the-art approaches. Specifically, phosphonium-based ionic liquids have been extensively investigated for the extraction and separation of platinum group metals. In this paper, we present the extraction capacity of several phosphonium-based ionic liquids for platinum group metals from model deep eutectic solvent-based acidic solutions. The most promising candidates, P_66614_Cl and P_66614_B2EHP, which exhibited the ability to extract Pt, Pd, and Rh quantitively from a mixed model solution, were additionally evaluated for their capacity to recover these metals from a spent car catalyst previously leached into a choline-based deep eutectic solvent. Specifically, P_66614_Cl afforded extraction of the three target precious metals from the leachate, while their partial separation from the interfering Al was also achieved since a significant amount (approx. 80%) remained in the leachate.

## 1. Introduction

The platinum group metals (PGMs) comprise the transition elements ruthenium, rhodium, palladium, osmium, iridium, and platinum. In particular, Pt, Pd, and Rh have the most crucial economic impact of all PGMs; they are employed in a wide range of applications and have become indispensable to various industrial processes. Owing to their distinctive characteristics—namely, high melting points and catalytic activity, low vapor pressure, and resistance to corrosion, they constitute essential components of electrical and electronic equipment, fuel cells, dental materials, medication, jewelry, and chemical and petroleum production. The most prominent PGM consumer is the automotive industry, where PGMs are used in vehicle catalysts to reduce the negative environmental impact of the produced exhaust gases [1,2]. Specifically, in 2021, the automotive industry demand worldwide for Pt was 39.4%, for Pd, 84.7%, and for Rh, 90.6%. Global jewelry manufacturing claimed 24.3% of Pt, while the second contenders for Pd were electronics with 5.9% and chemical production with 5.8% [3].

In the past few years, there is a steadily increasing demand for PGMs that exceeds the attainable supply, and according to future projections, this trend is not expected to change. Their natural abundance is low and their natural resources are globally limited and localized (mainly encountered in South Africa and Russia) [4]. At the same time, PGMs are largely irreplaceable in industrial applications, and according to geologists, the discovery of new resources is highly unlikely [5]. Although the current PGM resources are expected to last for a few decades longer, PGM recycling is of paramount importance since it signifies conservation of the already limited primary resources. Additionally, efficient recycling practices can minimize the exceedingly expensive mining operations of the continuously depleted deposits, consequently creating stability in the PGM market price [6]. Apart from the financial aspect, the negative environmental impact and health hazards associated with mining processes cannot be overlooked [7]. 

The state-of-the-art approaches for PGM recovery from secondary materials are based on hydrometallurgy and pyrometallurgy. Nevertheless, these processes have a number of drawbacks mainly associated with negative environmental impact and potential health risks. Hydrometallurgy relies on the use of strongly acidic (e.g., aqua regia, HCl, HNO_3_, H_2_SO_4_) or alkaline solutions (e.g., NaOH, cyanides) that contain toxic and easily decomposed components, which can generate dangerous and toxic gaseous compounds. Severe conditions of pressure (up to 6000 kPa) and temperature (up to 250 °C) are usually required, while the necessary thermal pre-treatment step prior to PGM leaching employs temperatures up to 800 °C. Mixing of the secondary material with oxides, reducing agents, and additives is performed prior to the smelting step in pyrometallurgical processes, where the employed temperatures are even higher (above 1000 °C), rendering these processes expensive and insufficient from an energy consumption perspective [8,9].

Alternative processes for the recovery of PGMs from spent materials are actively investigated by the scientific community. Magnetic recovery of PGMs from spent car catalysts has been reported [10]. The surface of the catalyst is coated with a ferromagnetic metal, i.e., Ni, which forms alloys with PGMs at a high temperature (800 °C). Following Ni deposition, the catalyst is pulverized, and the ferromagnetic properties on Ni are exploited to achieve separation of the metals from the ceramic material powder with the aid of a neodymium magnet (Nd-Fe-B alloy). The PGMs can be recovered from the Ni alloy either by direct dissolution of the magnetic powder in an acidic medium or by a pyrometallurgical approach where Ni serves as the collector metal [10]. The effect of hydrodynamic cavitation proved useful in enriching spent car catalyst powder in Pt and Pd, which were removed from the cordierite substrate due to the mechanical energy generated by the cavitation bubbles. Subsequent treatment of the sample with sono-electrochemistry accelerates the mass-transfer rate encountered in conventional electrochemistry, thereby affording higher recovery efficiency (approx. 32% higher) at reduced rates (1 h versus 24 h). Aside from the speed and low environmental impact of this approach, the price is a very attractive feature, i.e., 8EUR/g of metal, which is 5 times lower than the current market prices [11].

Ionic-liquid-based extraction of PGMs from spent car catalysts or other industrial waste materials is also increasingly investigated as an alternative approach to the traditionally employed hydrometallurgical processes [12]. Similarly, deep eutectic solvents (DESs), which are considered to be ionic liquid analogs [13], have also been reported in the recovery of precious metals, such as gold [14]. The distinctive and unique properties that characterize ionic liquids (ILs), such as negligible vapor pressure, wide solubility range, tunable nature, and simple variation in their building anions and cations and/or addition of selected functionalities that allow their design with a specific goal in mind, have rendered them suitable for a wide range of metal extractions and separations [15,16]. With respect to DESs, their low toxicity and volatility, simple property tuning by variation of the constituting components or their mixing ratios, as well as their biodegradability and relatively low cost [17], are typical characteristics that have rendered them viable candidates for alternative solvents in extractive metallurgy [18].

A considerable number of publications have dealt with the extraction of PGMs [19,20,21], as well as various other metals [22] and rare earth elements [23] with the aid of various types of ILs. Interestingly, in all cases in which ILs are employed, energy-efficient conditions are set, with temperatures close to room temperature and significantly faster extraction times.

The recovery of Pd and Rh from the HNO_3_ solution used in the treatment of spent nuclear waste with the aid of Aliquat 336 ([N_8881_]Cl) was already reported in 1968 [24]. Aliquat 336 N ([N_8881_][NO_3_]) [25] and S ([N_8881_][HSO_4_]) [26] have also been employed for the recovery of Pd and Pt, respectively. Interestingly enough, it has been proven that the selection of the organic solvent that serves as a diluent for Aliquat has a decisive impact on the distribution of the target metal [27]. Mixtures of imidazolium ILs have demonstrated high selectivity toward Pt from multi-element solutions [28], while their functionalization dramatically increases the selectivity for Pd, which is quantitatively extracted from multi-element mixtures [29]. Phosphonium-based ILs have already been applied in PGM recovery and separation from model solutions [26,30,31], and car catalyst acidic liquor [32,33]. The hydrophobic ILs based on phosphonium cations are thermally stable and, therefore, suitable for applications in a wide temperature range. Furthermore, their hydrophobic nature allows direct stripping of the PGMs from the phosphonium-IL phase [34].

In our previous studies, we investigated the recovery of PGMs from spent car catalysts with DESs. Complete recovery of Pt and Pd and a significant amount of Rh could be recovered under mild extraction conditions, i.e., low temperature and atmospheric pressure, in a straightforward and short process [35]; however, a number of accompanying elements, mostly Al, Fe, and Zn were co-extracted.

Herein, we expand this study to a liquid–liquid (L–L) extraction with ILs, aiming for separation and enrichment of PGMs. The extraction capacity for PGMs from their model DES-based solutions was studied with phosphonium-based ILs with various anions, before the ultimate application to a DES leachate of a spent car catalyst.

## 2. Discussion

### 2.1. Leaching of PGMs with Deep Eutectic Solvent

Prior to the separation experiments reported herein, a leaching procedure of PGMs from milled spent car catalyst material, with the aid of DESs, has already been developed in our research group [35]. The PGM-leaching capacity of various choline-based DES that was evaluated in our study was assessed at variable conditions; specifically, solid:liquid ratio, leaching temperature, leaching duration, amount and nature of the added oxidizing agent, presence or absence of additives, and dilution of the leaching medium with water.

The DES choline chloride/p-toluene sulfonic acid (p-TsOH) 1:1.8 eq was identified as the most suitable candidate and exhibited quantitative leachability for Pt and Pd (100%) and partial for Rh (approx. 50%), along with a number of other elements present in the car catalyst matrix. These results were afforded with the employed conditions of car catalyst:DES:HNO_3_ 65% 1:5:1, mixed for 4 h, at 80 °C (Table 1).

With this outcome in mind, the subsequent separation strategy was accordingly developed. The hydrophilic nature of the employed DES in the leaching step dictates the use of a hydrophobic IL in order to achieve the formation of a biphasic-liquid-based separation system. The developed strategy for PGM recovery is presented in Figure 1.

### 2.2. Evaluated Ionic Liquids for Liquid–Liquid Separation Studies

The evaluated ILs are based on the phosphonium cation with long alkyl chains, which is responsible for their hydrophobic nature (Figure 1). The ILs presented in this report were selected with certain key criteria in mind. First, hydrophobic nature was a required characteristic, in order to achieve the formation of a biphasic liquid system and subsequent separation of PGMs from accompanying elements by partitioning between the two liquid phases. High PGM selectivity was also an indispensable characteristic that would ensure the success of the separation process. The ionic liquids P_66614_Cl and P_66614_Doc were purchased, and the rest were synthesized in house. The detailed synthetic route for each IL is provided in the ESI.

We should draw attention here to the fact that despite the common misconception that ILs are safe and non-toxic, this is unfortunately not entirely true. The observed trend of increasing toxicity with increasing alkyl chain length is also applicable to phosphonium ILs. Of course, the building anion of the IL also has a significant impact on the toxicity level of the IL. Existing data for P_66614_-based ILs indicate that the C6 alkyl chains indeed lead to phosphonium ILs that are more toxic than their C4 counterparts. Additionally, exposure data of *Vibrio fischeri* to P_66614_Cl clearly demonstrate that this particular IL is slightly toxic, while it should be noted that toxicity increases with exposure time (EC_50, 15 min_ = 7.10 mg/L, EC_50, 30 min_ = 2.95 mg/L) [36]. With regard to financial considerations, a noteworthy review by Passos et al. can provide further insight into this topic [37].

### 2.3. Preliminary Liquid–Liquid Separation Experiments

The performance of the selected phosphonium ILs was evaluated with the aid of a PGM model solution based on the DES used in the developed leaching process; 1 part aqueous mixed PGM solution:5 parts DES:1 part HNO_3_ were all mixed, together with the aim of creating a model PGM solution that would most closely resemble the DES leachate. The reference to the PGM model solution throughout this paper will imply this solution (see ESI for details).

The selected phosphonium-based ILs were evaluated for their performance on PGM extraction from the car catalyst leachate. Due to the high viscosity of the involved ILs, we decided to work with diluted ILs and chose solutions of phosphonium IL in *n*-heptane in a 1:1 (*w/w*) ratio, to initially investigate the effect of different ILs. As standard conditions, RT and DES to phosphonium IL in n-heptane on a 1:1 ratio were preselected and applied in all experiments. In all cases, the extraction of Rh to the hydrophobic IL phase was significantly higher compared to Pt and Pd. The results of the systems tested for L–L separation are presented in Figure 2.

For the extraction of the PGMs from the DES phase to the phosphonium IL phase, an anion-exchange mechanism is assumed. The most commonly encountered species for Pt, Pd, and Rh in Cl-based solutions are PtCl_4_^2−^, PtCl_6_^2−^, PdCl_4_^2−^, PdCl_6_^2−^, [RhCl_5_(H_2_O)]^2−^, and RhCl_6_^3−^. The following extraction mechanisms are proposed; however, it should be noted that further studies are required for verification that are beyond the scope of this manuscript.:2P_66614_^+^X^−^*_(org)_* + PtCl_4_^2−^*_(aq)_* ⇌ [P_66614_][PtCl_4_]*_(org)_* + 2Cl^−^*_(aq)_*
2P_66614_^+^X^−^*_(org)_* + PtCl_6_^2−^*_(aq)_* ⇌ [P_66614_][PtCl_6_]*_(org)_* + 2Cl^−^*_(aq)_*
2P_66614_^+^X^−^*_(org)_* + PdCl_4_^2−^*_(aq)_* ⇌ [P_66614_][PdCl_4_]*_(org)_* + 2Cl^−^*_(aq)_*
2P_66614_^+^X^−^*_(org)_* + PdCl_6_^2−^*_(aq)_* ⇌ [P_66614_][PdCl_6_]*_(org)_* + 2Cl^−^*_(aq)_*

Since [RhCl_5_(H_2_O)]^2−^ is believed to be the most labile species, it is most probably the extractable one. The geometric configuration of RhCl_6_^3−^ prevents it from being extracted. The extraction mechanism of Rh is assumed to be the following.
2P_66614_^+^X^−^*_(org)_* + [RhCl_5_(H_2_O)]^2−^*_(aq)_* ⇌ [P_66614_]_2_[RhCl_5_(H_2_O)]*_(org)_* + 2Cl^−^*_(aq)_*
where X is the IL anion and *(org)* and *(aq)* denote the presence of each compound in the organic or aqueous phase, respectively.

### 2.4. Effect of Dilution of the Hydrophilic DES Phase

Along with the model PGM solution in DES, a 1:1 and 1:5 dilution with H_2_O were also evaluated. Additionally, the effect of a 1:1 dilution with 1 M NaCl was tested, since it has been previously reported in the literature that it can enhance PGM extraction [38,39]. The respective extraction efficiencies to the hydrophobic phosphonium IL phase are presented in Figure 3.

The dilution of the PGM-rich hydrophilic phase (DES) had a significant impact on the extraction efficiencies of all three PGMs to the phosphonium ILs. Evidently, dilution of the hydrophilic phase is necessary in order to achieve maximum extraction of the PGMs to the phosphonium IL. This cannot be attributed to the lowering of the concentration brought about through dilution but rather to the decrease in viscosity or/and formation of PGM species that are easier to extract. Application of the same experimental conditions in the real leachate, which has a fourfold concentration of Pt and sixfold concentration of Pd, compared with the model solution, proves that addition of water indeed enhances extraction, which arises from an effect other than the lowering of the concentration; thus, at least 1:1 dilution with water is necessary to achieve quantitative yield. Additionally, the dilution level does not seem to play a role in the extraction yield, except in the case of P_66614_Doc where higher dilution with H_2_O clearly favors the extraction of both Pt and Pd. Contrary to reported data, the addition of NaCl does not seem to further improve the extraction efficiencies; therefore, it was not used in subsequent optimization steps.

### 2.5. Effect of Dilution of the Hydrophobic Phosphonium IL Phase

For this set of experiments, the same phosphonium ILs mentioned in Section 2.4 were also evaluated. The separation experiments were performed with 50% (*w/w*) phosphonium ILs in *n*-heptane in order to reduce viscosity and facilitate their handling and mixing in the formed biphasic liquid system. Nevertheless, the impact of the pure phosphonium ILs on the extraction efficiency was also considered.

Since the optimization parameters presented previously demonstrated that the dilution of the hydrophilic phase was essential in order to obtain the maximum extraction efficiencies, all the subsequent optimization steps were performed with a 1:1 dilution with H_2_O of the PGM model system prior to separation. Therefore, from here on, whenever the PGM model system is discussed in the separation experiments, the 50% diluted version is implied.

In the experiments presented below, the PGM model system was mixed on a 1:1 ratio with the pure phosphonium ILs. The obtained extraction efficiencies are presented in Figure 4.

It is evident that employing pure phosphonium ILs in the separation step does not allow complete extraction of Rh, with the exception of P_66614_Doc. Therefore, in the subsequent optimization steps 50% (*w/w*) phosphonium ILs in *n*-heptane were employed, since the dilution of the phosphonium IL is a more economical approach than employing it in its pure form.

### 2.6. Effect of Acidity of Hydrophilic IL Phase

It has been reported in the literature that modification of PGM solution acidity by HCl addition allows effective separation of Pt and Pd from Rh [40,41,42]. This dependency of the extraction of Rh on the HCl concentration can be attributed to the presence of different Rh chlorocomplexes at different concentrations and their different extractability [43]. This approach was evaluated as a possible strategy for the separation of Rh from Pt and Pd. The concentration of HCl in the model PGM solution was adjusted to 3.7 M and 9.7 M. The extraction efficiencies in the respective acidities are presented in Figure 5. An increase in the HCl concentration from the original 1.0 M to 3.7 M and 9.7 M in the hydrophilic IL phase resulted in the complete separation of Rh from Pt and Pd, with all three different phosphonium ILs.

### 2.7. Application of Optimized Separation System to Authentic Car Catalyst Leachate

Based on the optimization experiments performed with the model PGM solution, the optimum conditions that allow for maximum extraction of the PGM to the phosphonium IL are the following; 1:1 mixture of DES phase (without HCl addition) diluted 50% (*w/w*) with water and phosphonium IL phase diluted 50% (*w/w*) with *n*-heptane, mixing time 2 h, at RT. Additionally, P_66614_Cl was further evaluated because it has higher extraction efficiencies for all three PGMs, and it is commercially available, as well as P_66614_B2EHP, because it demonstrated the capability to separate Pt and Pd from Rh in the model solution. These conditions and these 2 ILs were employed in a real car catalyst leachate.

Interestingly enough, contrary to the observations in the model solutions, when real leachate was employed, quantitative extraction of all three PGMs to both P_66614_Cl and P_66614_B2EHP was obtained. Additionally, unlike in the case of the model PGM solution, where Pt and Pd partitioned to the phosphonium IL and Rh remained in the DES phase, adjustment of the HCl concentration to higher levels—namely, 3.7 M and 9.7 M, had no effect on the separation of Rh from Pt and Pd, with both evaluated phosphonium ILs affording quantitative extraction for all three PGMs. On the contrary, regardless of the HCl concentration, quantitative extraction of Rh to both phosphonium ILs along with Pt and Pd was observed when the DES-based leachate was diluted with H_2_O prior to L–L separation, indicating that complete extraction of the metals from the DES to the hydrophobic IL can only be achieved by dilution with H_2_O, which is in agreement with what has been shown for the model solutions. Since both phosphonium ILs demonstrated similar PGM extraction from the real leachate (ESI, Appendix A), P_66614_Cl was employed in further experiments, as it is commercially available.

Subsequently, the performance of P_66614_Cl in the separation of PGMs from co-extracted interfering elements was evaluated. The quantification was based on multi-element analysis of the DES leachate before and after the separation step, and the results are presented in Figure 6.

The car catalyst comprised a multi-elemental matrix with Al, Fe, and Zn dominating. It has already been demonstrated that the DES, apart from achieving high extraction efficiencies for PGMs, extracts significant amounts of interfering metals. It is thus the task of the L–L separation step to target their removal. The developed L–L separation process allowed the enrichment of the hydrophobic IL phase with the target PGMs, while at the same time, there was almost complete removal of Al, which comprised by far the most significant interference, as well as a reduction in other interferences, i.e., Cu, Ni, Sr. In terms of extraction efficiencies to the hydrophobic IL phase, the following percentages correspond to each element (%): Pt 100, Pd 100, Rh 100, Al 20, Fe 100, Zn 98, Pb 91, Cu 56, Sr 80, and Ni 52. The validity of the results presented was verified by standard addition experiments.

### 2.8. Recycling of Hydrophilic IL after Liquid–Liquid Separation

The possibility of recycling and reusing the DES for new leaching cycles after separation was investigated. The PGM-rich leachate obtained from a leaching cycle was subsequently subjected to separation during which all PGMs were extracted to the phosphonium IL, thus leaving the DES free from PGMs. The PGM-free DES was recovered and re-used for subsequent leaching cycles. The goal of these experiments was to evaluate the capacity of the recycled DES to effectively leach PGMs and its stability throughout this process.

The recycling experiments were performed in four consecutive cycles and repeated for verification of the results. The number of leaching and separation cycles, as well as the respective PGM extraction, are presented in Figure 7. The recovery of the DES in each cycle was in the range of 50–60% of the input; the DES loss can be attributed partly to its viscosity and to handling losses during the experimental process, which are more severe in the small scale at which these experiments were performed.

No loss was observed in the leaching efficiency of the DES in the consecutive cycles performed; however, there was a considerable loss of the DES.

This can be attributed to physical losses related to high viscosity after the leachate centrifugation, but also to the migration of pTsOH from the DES P_66614_Cl in the separation step, which was verified via ^1^H-NMR spectroscopic data (ESI, Appendix A). Additionally, there was a modification in the appearance of the DES after separation—namely, the color changed from dark pink to yellow.

## 3. Materials and Methods

### 3.1. Chemicals

All reagents employed in the method development were of analytical grade. Concentrated hydrochloric acid 37% and nitric acid 65% were purchased from Merck, Germany. Stock solutions of Pt, Pd, and Rh 1000 ppm in 5% HCl were obtained from Fluka, Germany and used for the preparation of PGM model solutions and calibration standard solutions. P_66614_Cl was purchased from Iolitec, Germany, and P_66614_DOP from Sigma-Aldrich, Germany. The rest of the ionic liquids were synthesized in-house (detailed process provided in the ESI). The ^1^H-, ^13^C, and ^31^P-NMR spectra of the synthesized ILs were recorded from DMSO-d_6_ solutions on a Bruker Avance UltraShield 400 (400 MHz) spectrometer. The compounds used to provide the anion to the synthesized ionic liquids were all purchased from Sigma-Aldrich, Germany. High purity water was supplied by an Easipure water system (Thermo, USA, conductivity 18 MΩ·cm^−1^).

The car catalyst material employed in this work was provided by Monolithos Ltd. (Athens, Greece). The grinding size of the provided catalyst powder was <0.16 mm.

### 3.2. Instruments

The separation of the car catalyst material from the leaching solution was performed via centrifugation in a Microstar 12 tabletop centrifuge (VWR, Germany).

The leaching efficiencies of the PGMs in all systems before and after liquid–liquid separation were quantified with inductively coupled plasma–optical emission spectroscopy (ICP–OES) with appropriate sample dilution and matrix matching to accommodate for the high carbon content of the DES in the case of the leachates (1% EtOH in 5% HCl). The measurements were performed using a radial ICP–OES (Thermo iCAP 6500, Thermo Scientific, Waltham, MA, USA). A sample introduction kit consisting of a parallel path nebulizer (PEEK Mira Mist, Thermo Scientific, Ottawa, ON, Canada), a gas cyclonic spray chamber with a riser tube, and a torch injector tube with a 2 mm inner diameter was used.

^1^H-, ^13^C- and ^31^P-NMR spectra were recorded from CDCl_3_ and DMSO-*d*_6_ solutions on a Bruker AC 200 (200 MHz) or Bruker Avance UltraShield 400 (400 MHz) spectrometer. Chemical shifts (δ) were reported in ppm using tetramethylsilane as internal standard, and coupling constants (*J*) were given in Hertz (Hz). The following abbreviations were used to explain the multiplicities; s = singlet, d = doublet, t = triplet, q = quartet, quin = quintet, sext = sextet, m = multiplet.

### 3.3. Methods

Five replicas of each sample were prepared, and each replica was measured five times. Blank solutions with a composition identical to the diluent were used for the determination of limits of detection and quantification. Indium was used as an internal standard, and the output signals were corrected using the emission line at 230.606 nm.

The quantification of PGMs in the leachate before and after liquid–liquid separation was conducted within a 24 h frame since the completion of the leaching and separation process. Background corrected emission signals in ICP–OES were recorded and processed using Qtegra 2.10 software (Thermo Scientific, USA).

The stability of the hydrophilic ionic liquid (Choline Cl/pTsOH) and the selected hydrophobic ionic liquid (P_66614_Cl) after L–L separation was evaluated qualitatively via NMR spectroscopy.

### 3.4. Liquid–Liquid Separation Process

The model PGM solutions contained 48 mg/L (ppm) of each individual metal (Pt, Pd, Rh) in 1.0 M HCl, which was mixed with choline chloride/pTsOH and 65% HNO_3_, as already discussed. The solutions were freshly prepared on the day that the experiments were conducted. The spent car catalyst leachate contained 162 mg/L Pt, 265 mg/L Pd, and 23 mg/L Rh.

For the L–L separation, (0.2000 ± 0.0050) g of PGM acidic model solution (or DES leachate) was mixed with (0.2000 ± 0.0050) mL H_2_O to create the hydrophilic phase. For the hydrophobic phase, (0.2000 ± 0.0050) g phosphonium IL was mixed with (0.2000 ± 0.0050) g *n*-heptane (the addition of *n*-heptane facilitates the handling and mixing of the formed biphasic system by lowering the viscosity of the hydrophobic IL). The 2 phases were mixed on a 1:1 (*w/w*) ratio and stirred in a sealed glass vial for 2 h, at RT, at a stirring speed of 300 rpm. Upon completion of the stirring time, the mixture was placed in an Eppendorf tube and subsequently centrifuged for 5 min at 15,000 rpm (small scale) or 45 min at 15,000 rpm (upscaling) for fast and efficient separation of the 2 phases. After the centrifugation, the aqueous acidic phase was recovered and appropriately diluted for PGM quantification by ICP–OES.

The extraction efficiency from the hydrophilic to the hydrophobic IL was calculated by subtracting the measured concentrations before and after separation, according to the following equation:$$\%\;extraction\;efficiency=\frac{\mu g\;of\;M\;in\;input\;hydrophilic\;phase-\mu g\;of\;M\;in\;output\;hydrophilic\;phase}{\mu g\;of\;m\;in\;input\;hydrophilic\;phase}\times 100$$
where [M] = metal.

## 4. Conclusions

A number of phosphonium-based ILs were evaluated for their potential in extracting PGMs from DES, as well as separating them from accompanying elements. The impact of different parameters on the outcome of the L–L separation efficiency of the systems was assessed. The effects of the dilution of the hydrophobic and hydrophilic phase and the effect of modification of the acidity or the addition of salt to the hydrophilic phase were evaluated, and the ideal combination of the parameters that yielded the highest extraction efficiency of the PGMs to the phosphonium extractant was selected.

Several systems yielded high extraction efficiencies for all 3 PGMs; however, only P_66614_Cl and P_66614_B2EHP proved to be suitable candidates for complete extraction of all three targeted PGMs from the DES. Additionally, the dilution of both the DES and the phosphonium IL was deemed necessary in order to obtain quantitative extraction of the three PGMs. The phosphonium IL P_66614_Cl was the optimum extractant in terms of extraction efficiency but also commercial availability, which means a reduction in time-consuming synthetic efforts.

The possibility of separating Pt and Pd from Rh was investigated based on literature reports that stated that acidity is key to achieving this separation. Experiments were conducted with two different HCl concentrations, and the observations agreed with literature data in the case in which a PGM model system was employed. Nevertheless, the same approach had no effect on the L–L separation behavior of PGMs in the DES-based leachate; there was complete extraction of all three PGMs in the hydrophobic phase regardless of the HCl concentration.

The idea of recycling and reusing the hydrophilic IL after an entire leaching/separation cycle was investigated as a means of creating an approach that would be more financially sound and environmentally compatible in terms of generated waste. The DES proved to be equally efficient in the PGM extraction after recycling; however, the pTsOH component migrated to the phosphonium IL during separation, and therefore, redosing of the IL before a new leaching cycle was required. The phosphonium IL remained stable during separation; however, it was enriched with pTsOH migrating from the DES leachate.

The proposed method proves the feasibility of employing phosphonium-based ILs for PGM recovery, and it is envisioned that this approach can be employed in other end-of-life materials. Nevertheless, it should be pointed out that since there is a variety of spent materials with different PGM concentrations, adjustment of the developed conditions to the target material will most probably be necessary to accommodate complete PGM recovery.

## Data Availability

The data presented in this study are available on request from the corresponding author.

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
