# Peer review of "A Combined Deep Eutectic Solvent–Ionic Liquid Process for the Extraction and Separation of Platinum Group Metals (Pt, Pd, Rh)"

_molecules, 2021, doi:10.3390/molecules26237204_

Round 1

Reviewer 1 Report

  1. This is a research paper, so the abstract should include some of the research findings. At this stage, it simply feels like a paragraph of the introduction.
  2. Section 3, materials and methods need further classification. Chemicals and instruments should be written under separate headings. Moreover, methods should also be shifted under proper headings. Liquid-liquid extraction should be written with more clarity.
  3. The following paragraph is repeated three times (It can be found in sections 2.2, 2.3, 2.4). It seems an effort to populate the paper or authors did not bother to carefully read the paper before submission.

“The evaluated ILs are based on the phosphonium cation with long alkyl chains, which is responsible for their hydrophobic nature (Figure 2). The ILs presented in this report were selected with certain key criteria in mind. First, hydrophobic nature was a required characteristic, in order to achieve formation of a biphasic liquid system and subsequent separation of PGMs from accompanying elements by partitioning between the two liquid phases. High PGM selectivity was also an indispensable characteristic that would ensure the success of the separation process. The ionic liquids P66614Cl and P66614Doc were purchased and the rest were synthesized in house. The detailed synthetic route for each IL is provided in the ESI.”

  1. Which IL was selected based on the effect of dilution of the hydrophilic DES phase?
  2. Why only 1:1 ratio of IL with n-heptane was studied? What about other low and high dilutions?
  3. When almost 100% extraction was achieved by studying the effect of dilution of the hydrophilic DES phase, what is the significance of studying “Effect of dilution of the hydrophobic phosphonium IL phase” and “Effect of acidity of hydrophilic IL phase”.
  4. Where are the results of optimization of extraction time?
  5. The comparison of different DES and ILs has not been discussed adequately. It is important to study how the composition (alkyl chain length) can affect the extraction.
  6. Section 2.7 is poorly presented. If IL can extract all other metals along with the PGMs, then where is the selectivity of ILs for PGMs?
  7. The manuscript lacks organization, presentation, and clarity. The introduction section is organized and well written.

Reviewer 2 Report

I have reviewed article molecules-1471259, "A Combined Deep Eutectic Solvent-Ionic Liquid Process for the Extraction and Separation of Platinum Group Metals (Pt, Pd, Rh)" by Olga Lanaridi, Sonja Platzer, Winfried Nischkauer, Andreas Limbeck, Michael Schnürch and Katharina Bica-Schröder.

Comments and suggestions which authors may find useful in upgrading manuscript are the following:

  1. Introduction: different types of extractants of PGM should be short described. Not only phosphonium ILs.
  2. Title of a Chapter "Discussion" should be replaced by "Results and discussion".
  3. The reaction equations of PGM with the used extractants should be added to the manuscript.
  4. Chapter “2.6. Effect of acidity of hydrophilic IL phase” - Was HCl co-extracted with PGM? If yes, it could be reason, why extraction efficiency of Pd(II) (Figure 4, right) decreases?
  5. Has a stripping of platinum group metal from loaded organic phases (P66614Cl) been performed?

Reviewer 3 Report

The article is of very good quality with minor improvements that should be made before publication.

  1. Line 32-34. This is a qualitative phrase but it can be quantitative. For example, what is the autocatalyst ratio (normalized to the total gross demand) of Pt and Pd?
    pg. 38 from https://matthey.com/-/media/files/pgm-market-report/jm-pgm-market-report-may-2021.pdf
  2. Line 55-59. A phrase should be included here about the recent developments in the field.
    https://www.nature.com/articles/ncomms13164 
    https://www.sciencedirect.com/science/article/pii/S1350417720317089
  3. Table 1. I believe that Pt, Pd and Rh are missing a "+" sign or latin numbers.
  4. Line 352-355. Please comment on the following.
    The Experimental Section writes a 1:1 catalyst to IL. There are tonnes of automotive catalysts. Are the authors proposing to use the same ammount of IL? What are the costs? What are the environmental and health impacts of IL? How much heptane?

Author Response

Please, see attachment.

Round 2

Reviewer 1 Report

The authors have tried to address my comments.